# Inferior Vena Cava (IVC) Resections without Reconstruction in Renal Tumors: Two Case Reports

**DOI:** 10.3390/diagnostics13101759

**Published:** 2023-05-16

**Authors:** Bogdan Moldovan, Victor S. Costache, Irina Modrigan, Felix Farcas, Eugeniu Banu, Vlad Untaru, Doly Stoica, Madalina Crisan, Andreea Popianas, Radu-Mihai Pisica, Calin-Cristian Tohatan, Iris-Iuliana Adam, Liliana Vecerzan

**Affiliations:** 1‘St. Constantin’ Hospital, 500299 Brasov, Romania; 2Department of Cardiovascular Surgery, “Titu Maiorescu” University, 031593 Bucharest, Romania; 3Sanador Hospital, 011031 Bucharest, Romania; 4Faculty of Medicine, “Lucian Blaga” University, 550169 Sibiu, Romania

**Keywords:** IVC resection, carcinoma, clear cell, thrombosis, long-term survival, renal tumors

## Abstract

(1) Background: We aim to present our experience with resection of the inferior vena cava (IVC) without reconstruction in two patients diagnosed with renal tumors. (2) Case Report: The first case was diagnosed with right renal vein sarcoma and the second case was diagnosed with clear cell renal carcinoma; both presented signs of invasion and thrombosis of the IVC at infrarenal and cruoric levels, along with the development of collateral circulation with the help of the paravertebral plexus. In both patients, en bloc right nephrectomy was performed along with the resection of the thrombosed IVC without further reconstruction. In the case of the patient with right vein sarcoma, preservation of the left renal and caval intrahepatic vein was possible, whilst in the second case diagnosed with clear cell renal carcinoma, the associated left renal thrombosis also enforced the resection of the left renal vein. (3) Discussion: Postoperative evolution was favorable in both cases and did not exhibit major complications. Antibiotic therapy, analgesics, and anticoagulant medication were administered at therapeutic doses after surgery in both cases. The histopathological examination of the surgical specimen confirmed the diagnoses of renal vein sarcoma in the first case and clear cell renal carcinoma in the second case. Surgical treatment and adjuvant chemotherapy prolonged survival for two years for the first case and for two months, up until this moment, for the second case. The survival of clear cell renal carcinoma is currently at two months. (4) Conclusions: The resection of the inferior vena cava, without subsequent reconstruction in cases presenting diffused distal thrombosis, can represent an alternative to IVC reconstruction, which might lead to a major ulterior risk of thrombosis. In some cases, this can result in long-term survival.

## 1. Background

Retroperitoneal tumors are a heterogeneous group of diseases; most of them are lesions. Renal carcinomas, retroperitoneal liposarcomas, leiomyosarcomas, and retroperitoneal metastatic lymph nodes are the most common forms of malignant diseases located within this area. Radical resection (R0) is the only potentially curative option, even when the prognosis could remain unfavorable. The involvement of the inferior vena cava is uncommon and needs major vascular resection in most cases. The reconstruction of the IVC after resection is still controversial and open for debate. The first inferior vena cava resection was performed by Thomas Starzl, who was also the surgeon of the first liver transplant (1967) [1,2]. The reconstruction of the IVC following an immediate resection is considered mandatory so that an organism can preserve its normal function. Complete IVC resections without subsequent reconstruction were performed for the first time in abdominal trauma patients, proving to be feasible and safe and exhibiting a lower complication rate [3]. Still, in the medical literature, there are a small number of cases that benefited from this intervention. In order to be able to perform an IVC reconstruction, it is essential to correctly perform input–output methods so that the following anastomosis can retain its viability. The tropism of renal tumors, which also present vascular invasion, is still incompletely elucidated. They frequently cause thrombosis in the renal vein because of their shorter length compared to the cava.

## 2. Case Report

The first case presents a 57-year-old female patient with no significant medical personal history. The symptomatology started a month earlier when the patient reported pain in the right lower limb, along with swelling, paresthesia, difficulty in walking, moderate dyspnea, and fatigue, all of which were experiences that motivated her to contact our cardiology service. Our laboratory investigations revealed altered coagulogram values, highlighting an increase in D-dimers. A venous echo Doppler was performed on the lower limbs, which showed left ileo-femoral deep vein thrombosis (DVT). Abdominal–pelvic computed tomography (CT) identified a voluminous right inter-reno-caval retroperitoneal mass, with the invasion of the right renal hilum, stage III renal hydronephrosis, and IVC thrombosis (Figure 1).

Subsequently, after surgical evaluation, it was decided that a right nephrectomy with an inferior vena cava resection without ulterior reconstruction was needed. Intraoperatively, thrombosis was identified, which determined inflammatory phenomena that did not allow for recanalization of the intrahepatic-reno-periumbilical vena cava. The IVC was obliquely sectioned, and the preservation of the left renal vein at its implantation site was carried out. Sectioning was performed at approximately 2 cm above the IVC’s bifurcation with the help of a 45 mm TEA stapler, along with the application of a clip on the right renal artery (Figure 2).

Postoperative evolution was favorable under the anticoagulant treatment administered in therapeutic doses with low-weight-molecular heparin, with ulterior indications of continuing oral treatments using apixaban. The patient’s initial discharge took place nine days after surgery but with a second admission two weeks later for a deep venous thrombosis located in the right lower limb. It was decided that the dose of anticoagulants should be increased for the acute condition along with a new imaging examination, which ulteriorly did not show significant postoperative changes.

The histopathological examination correlated with immunohistochemical (IHC) tests and advocated for low-grade leiomyosarcoma with neuroendocrine components and an apparent origin in the renal wall. The proliferation index, ki67, was 15%. NSE was positive with moderate intensity (++) on 100% of tumor cells (Figure 3).

The patient was discharged with recommendations of starting chemotherapy treatment targeted for retroperitoneal G2 leiomyosarcoma with the neuroendocrine component. The first chemotherapy cycle was initiated 10 weeks after surgery with carboplatin 5 AUC IV for 30 min on the first day, followed by etoposide at 100 mg per m^2^ on the first and third days and once every three weeks. At the end of the first cycle, grade I neutropenia was discovered, for which it was decided that the doses of carboplatin that start the second cycle should be decreased, along with the subsequent administration of the G-CSF granulocyte growth factor at 0.5 mu/kgc/day 24 h after the end of the cycle.

The second case presents a 45-year-old male patient who presented to our service for lumbar back pain, asthenia, fatigue, and functional problems in both lower limbs. An abdominal CT was performed, which revealed a giant tumor mass affecting the upper 2/3 of the right kidney, which was slightly increased in size whilst also presenting a tumorous invasion of the right renal vein and the proximal IVC along with the presence of multiple collateral circulation paths (Figure 4).

The initial histopathological examination was performed in another oncology institute and revealed G2 clear cell renal carcinoma. Following the surgical examination performed in our service, it was decided that an en bloc right nephrectomy and resection of the inferior vena cava without subsequent reconstruction were needed. During surgery, thrombectomy of the inferior retrohepatic caval vein and ligature of the left renal vein was carried out without the need for an ulterior reconstruction, ensuring a complete resection of the tumor and thrombus. As a final surgical step, para-aortocaval and inter-aortocaval lymphadenectomies were performed, further providing a clean R0 resection. Postoperatively, the patient presented chylous ascites, requiring the prolonged maintenance of drain tubes for one month after the surgery. Under anticoagulant treatments in therapeutic doses, the evolution of the patient was favorable, allowing his discharge at 10 days after surgery (Figure 5).

The histopathological examination correlated with the IHC tests and reconfirmed clear cell renal carcinoma: SUP G2 grade, localized in the right kidney, and pT3b N1 (1/17) Mx L1 V1 Pn0 R0 (Figure 6).

This case is being discussed in the oncology committee, and starting therapy with oral sunitinib once a day for 28 days with the repetition of the cycle every 6 weeks (4 weeks of treatment, followed by a 2-week break) was proposed.

The postoperative evolution in both cases was favorable. In the case of the first patient, approximately two weeks after the intervention, she presented a Class II Clavien right lower limb thrombosis, which was treated with intravenous anticoagulants, whilst in the case of the second patient, chylous ascites was highlighted, requiring the maintenance of drain tubes for one month after surgery. The survival rate in the case of the first patient was set at 20 months, and in the case of the second patient, the survival rate was set at 2 months.

The surgical algorithm was proposed by the multidisciplinary team of the St. Constantin Hospital Brasov, presenting various surgical options in the case of renal tumors that simultaneously present IVC thrombosis, and the algorithm can be used if there is no proof of metastatic disease. The algorithm was developed after the two successful IVC resections, which did not require an ulterior reconstruction (Figure 7).

## 3. Discussion

The invasion of the IVC by tumors located in the retroperitoneal space may require resection of the IVC with or without reconstruction. For lesions involving more than half of the circumference, resection is indicated. The reconstruction of the IVC after resection is still controversial, and the availability of information is limited.

Vicente et al. [4]—in a retrospective study carried out at the Sanchinarro University Hospital in Spain from 2005 to 2015, which included twenty patients with the surgical resection of retroperitoneal tumors associated with IVC resection—showed that more than half of the cases developed later complications, and death occurred in only one patient. In addition, only one case developed postoperative thrombosis 14 months after surgery. This study concluded that tumor surgical resection concurrent with that of IVC can be easy, but only in carefully selected cases.

The postoperative complications experienced by our patients were deemed to be deep vein thrombosis and chylous ascites. In a study conducted by Duty et al. [5] on six patients with retroperitoneal tumors associated with an IVC invasion who underwent an IVC resection without reconstruction, the study showed that the most frequent postoperative complications were pneumothorax, lower extremity compartment syndrome, right upper extremity brachial plexus stretch injury, respiratory failure, and atrial fibrillation. This study concluded that IVC resection without reconstruction was well tolerated by patients with massive retroperitoneal tumors because most patients had well-defined collateral circulation prior to surgery because of the pre-existing caval obstruction. One of the most important factors in the field of oncological surgery is the complete resection of the tumor as well as the complete excision of any affected major vascular structures, such as the IVC.

The survival rate in the case of our first patient was set at 20 months currently, whilst in the case of the second patient, it was set at 2 months. A study conducted by the British Association of Urological Surgeons (BAUS) [6] in 2021—on 28 patients presenting retroperitoneal tumors—highlighted the fact that 50% of patients that were operated on later developed complications, which include sepsis, pneumonia, congestive heart failure, acute renal failure, symptomatic deep vein thrombosis, and splenectomy. The median follow-up 21 months later states that six patients died; for two of them, cancer progressed to a metastatic stage, and 71.4% showed no further signs of progression. The conclusions of this case series study illustrate the experience gained in the IVC resection without reconstruction, and this was considered a safe procedure later.

Cocchi et al. [7] conducted a study in which they reported a series of three patients with retroperitoneal tumors that also had signs of IVC invasion. All patients underwent a radical R0 surgical resection associated with infrarenal IVC resection without further reconstruction. Based on preoperative and intraoperative imaging findings, one patient underwent right nephrectomy while another patient underwent left renal vein ligation without nephrectomy. No severe early or late postoperative complications related to the absence of flow in the IVC were observed. Resection without the reconstruction of the infrarenal IVC results in an acceptable morbidity rate, highlighting that risks related to the use of further prosthetic grafts can be fully avoided.

In a research study published in 2020, Ceccanti et al. [8,9] reported three cases of Wilms tumors, which also presented total vascular IVC occlusion due to a thrombus. Total nephrectomies were performed, along with the complete resections of the IVC, without subsequent reconstructions. All patients were asymptomatic and disease-free up to two years after surgery. The conclusion was that in patients with Wilms tumors that also had a thrombus that completely occluded the IVC, cavectomies without reconstruction were considered safe and well tolerated. In addition, en bloc nephrectomy with clear margins eliminated the need for postoperative radiotherapy. Mortality rates were considered negligible and showed promising optimal functional results [10].

According to the algorithm proposed by our team, in a retrospective analysis of 87 patients conducted between 1997 and 2008 at the Urology Department of the University of Miami—led by Ciancio et al. [11]—an algorithm for classifying patients according to the localization of the IVC thrombus was proposed. Level I presented patients with infrarenal thrombus localization, level II presented intrahepatic levels, whilst level III presented intrahepatic levels and level IV presented suprahepatic levels. The study concluded that total nephrectomy and IVC thrombectomy offered reasonable long-term survival. Moreover, the location of the thrombus should not be considered a single prognostic factor in these cases [12].

Horodyski et al. [13] further considered that an ulterior reconstruction of the inferior vena cava is not mandatory if adequate collateral circulation can be identified. Renal cell carcinomas (RCCs) do not usually extend to IVC levels; however, in some cases, when they do extend to IVC levels, they often present as tumor thrombus (TT). The surgical en bloc resections of the tumor and affected IVC portion are considered the first option for treatment due to the potential improvement in the oncological prognosis.

However, some authors still suppose that the reconstruction of the IVC after a cavectomy is mandatory. Baldrich et al. [14] claimed that reconstruction is not optional for malignant tumors because a replacement of the IVC with the help of various prosthetic grafts can help avoid a number of venous complications, and it can also improve the survival rate. The survival rate also depends on the tumor behavior and IVC invasion. Primary patch repairs showed a significantly improved outcome compared with a circumferential resection [15,16].

Another important aspect to be considered in these cases is represented by the pre- and postoperative imaging done with the help of computed tomography (CT) and ultrasound, showing the extent of the IVC invasion of the thrombus as well as of the tumorous formation.

The non-invasive evaluation of fluid responses appearing in the inferior vena cava was addressed by Luigi La Via et al. [17] in a study, where it was highlighted that diam-eter variations during ventilation can further influence its caliber measurements, yet still make it possible to have a correct assessment while using the coronal trans-hepatic approach.

Furthermore, one of the primary expressions of the caval thrombus presence in re-nal cell carcinoma (RCC) patients is given by the extensive wall invasion, which could alter the surgical outcome. Qui-Yang Li et al. [18] concluded in a study from 2019 that contrast-enhanced ultrasound (CEUS) could help in differentiating a bland thrombus from a tumor-induced one.

## 4. Conclusions

The involvement of the inferior vena cava in the context of oncological pathology is considered a criterium of inoperability, even more so when it is related to thrombotic vascular obstruction. Our multidisciplinary team wants to highlight the fact that even with a thrombus at the IVC level, which preserved functional blood flow, a safe R0 surgical resection of the tumorous formation can be performed with the help of a thrombectomy via a cavotomy followed by cavoraphy. This will further ensure a long-term disease-free survival rate.

## Figures and Tables

**Figure 1 diagnostics-13-01759-f001:**
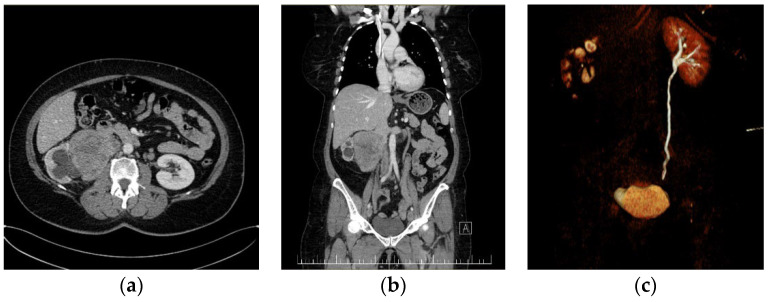
Preoperative CT imaging of the first patient: (**a**) giant right kidney—tumor with the invasion and thrombosis of the IVC. (**b**) Coronal CT image presenting a giant right kidney tumor with the invasion and thrombosis of the infrarenal IVC. Left suprarenal vein and left IVC are within normal ranges. (**c**) Coronal CT reconstruction in the excretion phase, which highlights the normal urinary excretion of the left kidney and its absence in the right kidney.

**Figure 2 diagnostics-13-01759-f002:**
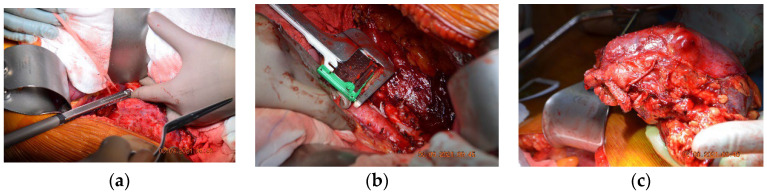
Intraoperative photos of the first case. (**a**) Suprarenal oblique IVC resection with a vascular stapler whilst conserving the left renal vein. (**b**) Inferior resection of IVC using the 45 mm TEA stapler. (**c**) The resected tumorous formation: right kidney and inferior vena cava.

**Figure 3 diagnostics-13-01759-f003:**
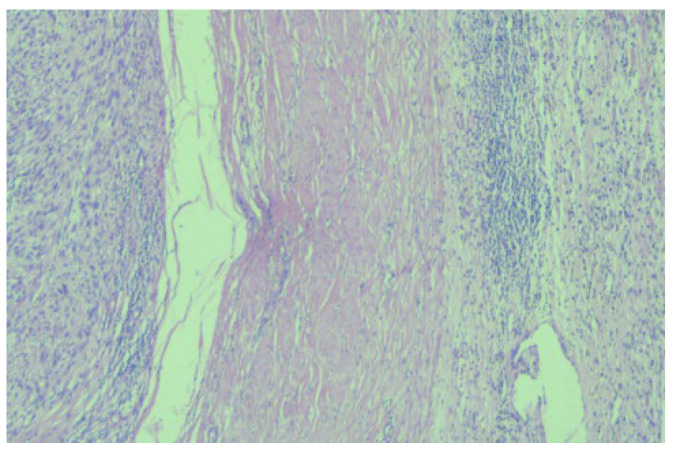
Microscopic image (×20) from the histopathological examination showing the presence of a tumorous thrombus in the lumen of the venous wall.

**Figure 4 diagnostics-13-01759-f004:**
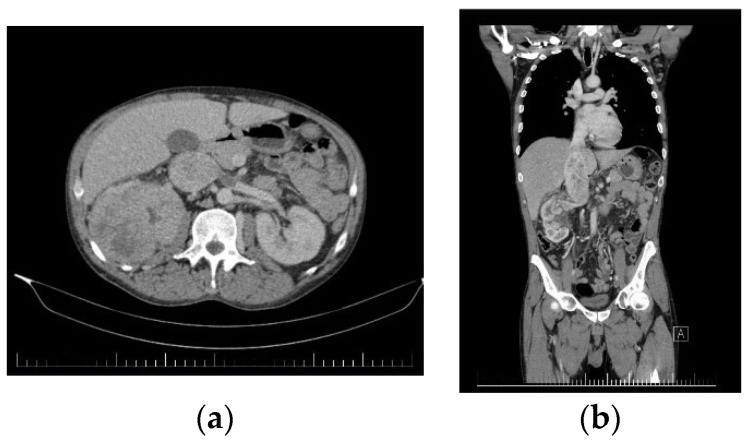
Preoperative CT images of the second patient. (**a**) Axial CT image showing a giant right kidney tumor along with the thrombosis of the IVC and last part of the renal vein. (**b**) Coronal CT image of the tumor with retrohepatic vena cava tumoral thrombosis and infrarenal vena cava cruoric thrombosis.

**Figure 5 diagnostics-13-01759-f005:**
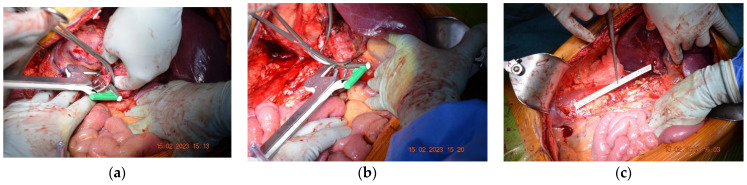
Perioperative images of the second case. (**a**) The resection of the IVC with a TEA stapler. (**b**) Left renal vein resection with a TEA stapler. (**c**) Final postoperative aspect.

**Figure 6 diagnostics-13-01759-f006:**
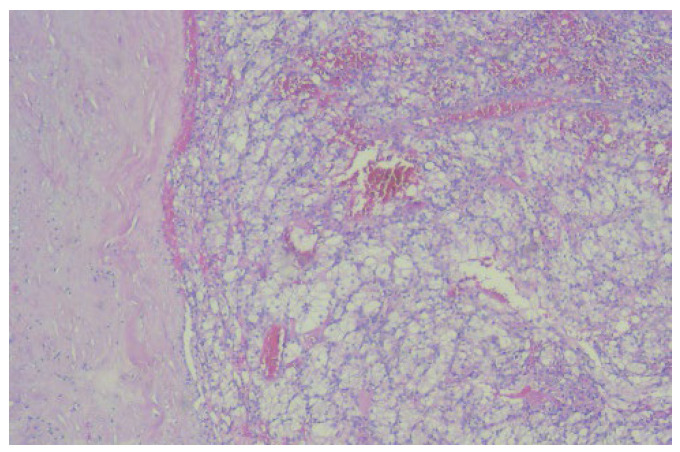
The microscopic image (×40) from the histopathological examination shows the presence of a tumorous thrombus in the lumen of the venous wall, which indicates the presence of clear cell renal carcinoma.

**Figure 7 diagnostics-13-01759-f007:**
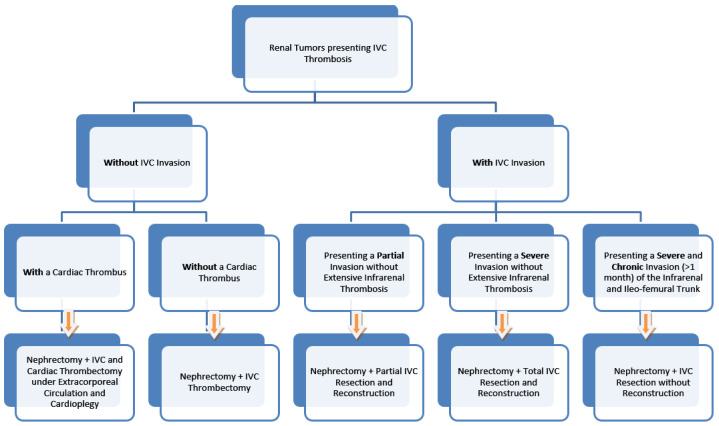
The proposed surgical algorithm.

## Data Availability

Not applicable.

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
