# Peer review of "Inferior Vena Cava (IVC) Resections without Reconstruction in Renal Tumors: Two Case Reports"

_diagnostics, 2023, doi:10.3390/diagnostics13101759_

Round 1
Reviewer 1 Report
I am thankful for giving me the chance to review the manuscript entitled “Inferior vena cava (IVC) resections without reconstruction in 2 renal tumors: Two Case Reports”. The topic is with novelty and the clinical significance is existed, but I have few concerns:
1. The format of the article was incorrect. In general, case report should follow CARE (Case Report) Guidelines and per journal style. You can refer to 2 case reports recently published in Diagnostics: Diagnostics 2023; 13 (8): 1368 and Diagnostics 2023; 13 (8): 1392.
2. The abstract also needs to be re-written.
3. There are many words in the manuscript should be modified. I suggest seeking for English editing and re-formatting. For examples in the Introduction:
(1) “…with most of them being “lesions” ”. The sentence is not clear.
(2) “Radical surgery (R0)” should be modified to “Radical resection (R0 resection)”.
(3) “… is also the author of …” should be modified to “… is also the surgeon of …”
(4) “a n immediate” should be modified to “an immediate”.
(5) “The involvement of the IVC” should be modified to “The involvement of the interior vena cava (IVC)”.
(6) “The reconstruction of interior vena cava” should be modified to “The reconstruction of IVC”.
(7) “… in abdominal trauma patients, proving … rate.” You should cite references here.
(8) “VCI” should be modified to “IVC”.
The author used a lot of uncommon words in medical presentations.
Reviewer 2 Report
I read with great interest the manuscript by Moldovan et al. on two case reports describing inferior vena cava resections without reconstruction for 2 renal tumors. The two cases are interesting and well reported. However, I have some minor issues to be addressed:
Introduction
- Line 41. There is a mistake when writing "an immediate resection".
- A description should be added on the importance of IVC evaluation with CT scan (doi: 10.1111/j.1365-2265.2011.04177.x) and ultrasound (doi: 10.4103/0028-3886.355107), both for surgery (doi: 10.1186/s40644-019-0265-x) and for hemodynamic assessment (doi: 10.1016/j.jcrc.2022.154108 - doi: 10.1186/s40635-023-00505-7). Please briefly discuss and add these 5 references.
- Line 42-44 "Complete IVC resections without subsequent reconstruction were performed for the first time in abdominal trauma patients". Please provide one or more references for this sentence.
Methods
- Line 53 Please describe the start of symptomatology compared to the day of surgery (i.e. 20 days before surgery)
Figure 7
- Was this algorithm proposed after these 2 cases or before? Please specify.
Some small corrections are required (as reported in my "comments" section.
Round 2
Reviewer 1 Report
The current structure of abstract, including aim, materials and methods, results, and conclusion is incorrect and should be modified to a case report format, including background, case report, discussion, and conclusion.
It is okay.
Author Response
We responded in the attached document.